# Potential miRNAs as Diagnostic Biomarkers for Differentiating Disease States in Ulcerative Colitis: A Systematic Review

**DOI:** 10.3390/ijms26146822

**Published:** 2025-07-16

**Authors:** Atta Ullah Khan, Pilar Chacon-Millan, Paola Stiuso

**Affiliations:** Department of Precision Medicine, University of Campania Luigi Vanvitelli, 80138 Napoli, Italy; attaullah.khan@unicampania.it (A.U.K.); pilar.chaconmillan@unicampania.it (P.C.-M.)

**Keywords:** ulcerative colitis (UC), microRNAs, active ulcerative colitis (aUC), inactive ulcerative colitis (iUC), biomarkers

## Abstract

Ulcerative colitis (UC) is a chronic inflammatory disease that affects the colon, triggering persistent inflammation and ulceration, resulting in a severe impact on patients’ quality of life. Currently, the standard diagnostic methods for UC include invasive procedures such as colonoscopy and the use of non-specific inflammatory markers like C-reactive protein, which can be inconvenient or painful and lack specificity. This underscores the need for non-invasive and highly specific biomarkers for UC. MicroRNAs (miRNAs) are small non-coding RNAs, typically 22 nucleotides in length, which are well described as gene expression regulators. Several studies have reported their differential expression in various pathological conditions, including UC. Due to their role in gene regulation and stability in biological fluids, miRNAs present a promising opportunity as biomarkers. This systematic review explores the potential use of miRNAs as diagnostic biomarkers to distinguish between active and inactive ulcerative colitis. Following PRISMA guidelines and based on inclusion and exclusion criteria, seven studies, encompassing a total of 514 participants (181 with active UC and 116 with inactive UC), were included. Multiple miRNAs exhibiting differential expression between active and inactive UC were identified. Most notably, miR-21, miR-126, miR-146b-5p, and miR-223 exhibited consistent upregulation in active UC, suggesting their potential as diagnostic biomarkers. Supporting these findings is the fact that these miRNAs are involved in inflammatory pathways, further highlighting their relevance to the pathogenesis of UC. This review emphasizes the need for further validation studies with larger cohorts to confirm the utility of miRNAs as diagnostic tools for UC disease activity differentiation, which could enhance non-invasive disease monitoring and inform therapeutic decision-making. Future research should also evaluate the prognostic potential of these miRNAs for predicting treatment responses and long-term disease outcomes.

## 1. Introduction

Ulcerative colitis (UC) is a chronic inflammatory disorder that affects the colon and rectum. It is distinguished by continuous inflammation without healthy areas between affected regions. This inflammation is confined to the rectum and large intestine, specifically targeting the mucosal layers of the epithelium [1]. A hallmark of UC is the formation of inflammation-induced ulcers in the colon barrier. Symptoms include rectal bleeding, abdominal pain (usually in the lower left quadrant), the need to defecate, and bloody diarrhea. Complications that typically arise include colonic issues such as toxic megacolons, perforations, and colitis-associated cancer. Extra-intestinal complications involve arthritis, primary sclerosing cholangitis (PSC), weight loss, anemia, and blood clots (thromboembolism), all of which significantly affect an individual’s quality of life. The diagnosis is based on medical history, along with radiographic, histological, and endoscopic examinations [1,2]. Treatment options typically include immunosuppressants, anti-inflammatories, biologics, and surgery for severe cases [3,4,5]. Although its precise etiology is uncertain, UC is thought to have a complex etiology that includes both environmental and genetic variables [6]. Specific data on the global prevalence and incidence of UC are limited. However, inflammatory bowel disease (IBD) in general has a high prevalence in developed or industrialized countries, possibly due to longer life spans with disability, while its incidence appears to be decreasing. Conversely, newly or rapidly industrializing countries are experiencing an increase in incidence. For instance, the annual prevalence of UC per 100,000 persons increased significantly from 5 in 2010 to 98 in 2019 in Japan and from 158 to 233 in the US during the same period. In the UK, as of December 2018, the prevalence and incidence of UC were reported to be 397 and 15.7 per 100,000 people, respectively, with age being a major factor influencing these trends [7,8,9,10]. Currently, UC diagnosis and monitoring rely on invasive methods like colonoscopy, alongside non-specific biomarkers such as C-reactive protein and erythrocyte sedimentation rate. While fecal calprotectin is more specific, there remains a need for improved biomarkers. 

Beyond these commonly used markers, several other biomarkers are employed in UC management, each with distinct limitations. Fecal lactoferrin and neutrophil elastase provide additional fecal-based inflammatory indicators, while serological markers including anti-neutrophil cytoplasmic antibodies (ANCAs) and anti-Saccharomyces cerevisiae antibodies (ASCAs) offer diagnostic utility but cannot differentiate disease activity states. Cytokine panels measuring TNF-α, IL-6, and IL-10 reflect inflammatory processes but are expensive and technically complex and lack disease specificity. Serum amyloid A and albumin levels provide general inflammatory information but suffer from poor specificity for UC activity. These existing biomarkers predominantly reflect downstream inflammatory consequences rather than the underlying molecular mechanisms driving disease pathogenesis, highlighting the need for more mechanistically relevant biomarkers.

MicroRNAs represent a paradigm shift in biomarker development, addressing key limitations of conventional inflammatory markers. Unlike proteins that reflect secondary inflammatory processes, miRNAs directly regulate gene expression at the post-transcriptional level, providing mechanistic insights into disease pathogenesis. Their remarkable stability in biological fluids surpasses that of proteins and messenger RNAs, with resistance to RNase degradation and temperature fluctuations that plague traditional biomarkers. The tissue-specific and disease-specific expression patterns of miRNAs offer enhanced specificity compared to broad inflammatory markers like C-reactive protein. Furthermore, their multiplexing potential allows the simultaneous assessment of multiple regulatory pathways using standard qPCR platforms available in routine clinical laboratories, potentially providing a comprehensive molecular signature of disease activity rather than single-parameter measurements.

It is essential to distinguish between diagnostic and prognostic biomarker applications in UC management to ensure appropriate interpretation of findings. Diagnostic biomarkers are utilized to identify the presence of disease or differentiate between disease states, such as distinguishing active from inactive UC, which represents the primary focus of this systematic review. Prognostic biomarkers, in contrast, predict future disease outcomes, treatment responses, or risk of complications such as hospitalization, surgery, or malignant transformation. While our systematic review addresses the diagnostic application of miRNAs for disease activity differentiation, the identified biomarkers may possess dual utility. Disease activity monitoring inherently provides prognostic information, as patients with persistently active disease face increased risks of complications and may require treatment intensification. Future studies specifically designed to evaluate the prognostic applications of these miRNAs, including the prediction of treatment response, disease progression, and long-term outcomes, would provide valuable complementary information for comprehensive UC management.

Therefore, this systematic review addresses the following specific research question: “Can miRNAs serve as reliable diagnostic biomarkers for differentiating active from inactive ulcerative colitis, and what are their performance characteristics compared to existing biomarkers?” Our primary objective is to systematically evaluate the differential expression patterns of miRNAs between active and inactive UC states, assess their diagnostic accuracy parameters where reported, and determine their clinical utility for non-invasive disease activity monitoring. Secondary objectives include identifying the most promising miRNA candidates for clinical translation, evaluating their mechanistic relevance to UC pathophysiology, and assessing the quality of evidence supporting their biomarker potential. This systematic approach follows the PICO framework to ensure a comprehensive evaluation of Population (UC patients), Intervention (miRNA measurement), Comparison (active vs. inactive disease states), and Outcome (diagnostic accuracy for disease activity differentiation).

## 2. Methods

### 2.1. Search Strategy and Eligibility Criteria

The search for this systematic review was conducted from inception to 4th September 2024, in line with Preferred Reporting Items for Systematic Reviews and Meta-Analyses (PRISMA) statement guidelines, and was registered on PROSPERO, with the registration No CRD42024597537 [11]. The databases utilized included PubMed, Embase, and Web of Science. The search terms used included (“mir”) OR (“microrna”) OR (“mirna”) AND (“biomarkers”) AND (“ulcerative colitis”) OR (“active ulcerative colitis”) OR (“inactive ulcerative colitis”). The detailed search and selection strategy is shown in Table 1 and Figure 1.

The screening of the identified records was performed by two independent investigators (P.C.M and A.U.K) by title and abstract first, and later, full-text screening was carried out. Studies fulfilling the pre-defined inclusion criteria utilizing the PICO approach were included as outlined in Table 2.

### 2.2. Quality Assessment and Critical Appraisal

The quality assessment of each individual study was carried out using the Newcastle–Ottawa Scale (NOS), which is based on three broad criteria, the first being the Selection of population being studied, second Comparability checks for variables other than what is being studied and how well these are taken care of, and lastly how accurate are the procedures carried out during these studies and if they are the same for both sets of populations under study. The assessment of all studies is presented in Table 3. The most frequently encountered risk of bias was that associated with the control of confounding factors (all studies) and attrition bias (all studies). By adding up all the individual criteria, we obtained the total score for each individual study, according to which five studies (71%) scored (≥56%) and were classified as “fair” quality with a “moderate risk of bias”, while two studies (29%) scored (77.77%) and were classified as “good” quality having a “low risk of bias”.

The quality assessment revealed several systematic limitations that affect the interpretation of findings. Beyond the identified risks of bias related to confounding factors and attrition, additional methodological concerns emerged. The small sample sizes across all studies (10–55 participants per group) limit statistical power and generalizability of the findings. The absence of power calculations in most studies suggests that studies may have been underpowered to detect clinically meaningful differences. Heterogeneity in disease duration, medication use, and demographic characteristics across studies introduces potential confounding variables that could influence miRNA expression patterns independently of disease activity. Sample collection and processing protocols varied significantly between studies, with different storage conditions, extraction methods, and normalization approaches potentially contributing to expression variability. The lack of standardized protocols for sample handling represents a critical gap that must be addressed for clinical translation. Additionally, the absence of validation cohorts in most studies limits the reliability of reported findings and highlights the preliminary nature of current evidence. These limitations underscore the need for larger, more rigorously designed validation studies with standardized protocols, appropriate power calculations, and comprehensive control for confounding variables before clinical implementation can be considered. 

### 2.3. Data Extraction

Two reviewers independently extracted data using a standardized form. The information collected included study characteristics (first author, year, country, design), participant details (sample size, age, sex, disease status), and miRNA measurement methods (specimen type, extraction and quantification platforms, normalization techniques). Key findings on differential miRNA expression between active and inactive ulcerative colitis were recorded, along with reported diagnostic implications. Discrepancies were resolved through discussion or third-party consultation.

### 2.4. Data Synthesis

A narrative synthesis was conducted to evaluate the diagnostic potential of miRNAs in distinguishing active from inactive ulcerative colitis. Studies were analyzed thematically, focusing on consistent miRNA expression patterns, direction of regulation, and association with disease activity markers. The findings were grouped by miRNA identity and clinical context to identify recurring biomarkers and methodological trends.

## 3. Results

Seven studies were included by following the search criteria in this systematic review. Studies included a total of 514 participants (active UC, 181; inactive UC, 116). Data extracted from each included study comprised study characteristics (authors, year, design), participant details (sample size, demographics), miRNA measurement methods, and key findings about miRNA expression patterns in each individual study in different disease states. This information is collected and presented in Table 4 and Table 5.

### 3.1. Study Characteristics and Methodological Diversity

The seven included studies demonstrated substantial heterogeneity in design and methodology (Table 4). Sample sizes varied considerably, ranging from 27 participants [14] to 132 participants [16], with most studies including relatively small cohorts that may limit statistical power. The diversity in sample types reflects the versatility of miRNA detection: three studies utilized tissue samples (colonic biopsies), two employed serum samples, one used fecal samples, and one analyzed blood samples. This methodological diversity, while demonstrating miRNA stability across biological matrices, may contribute to the variation in expression patterns and fold changes observed among the studies. Gender distribution was reported in four of the seven studies and appeared relatively balanced where documented. However, three studies [16,17,18] failed to report gender demographics, representing a potential source of bias. Disease duration varied significantly both within and between studies, ranging from 0.5 to 20.1 years, which may influence miRNA expression patterns and complicate cross-study comparisons. All studies employed quantitative PCR-based approaches for miRNA measurement, with some utilizing microarray screening for initial discovery, ensuring methodological consistency in quantification approaches.

### 3.2. miRNA Expression Patterns and Cross-Study Consistency

The analysis of miRNA expression patterns revealed several candidates with consistent differential expression between active and inactive UC (Table 5). miR-21 emerged as the most consistently upregulated miRNA in active UC, with [12] demonstrating a remarkable 14.7-fold increase compared to the healthy controls (*p* < 0.05), while [14] confirmed significant upregulation of miR-21-5p in active UC tissue samples. However, Ref. [15] reported contradictory findings, with miR-21 showing decreased expression in active UC serum compared to inactive UC serum, highlighting the importance of sample type in miRNA biomarker interpretation. miR-126 demonstrated the highest fold change in active UC, with an 18-fold increase reported by [12] in tissue samples. This substantial upregulation, combined with the lack of significant changes in inactive UC, suggests miR-126 may serve as a specific marker for active inflammatory states. miR-146b-5p showed consistent upregulation across serum-based studies, with [17] reporting significant overexpression in active UC compared to both inactive UC and healthy controls, supported by [18] despite their limited inactive UC cohort (n = 2). miR-223 expression patterns varied by sample type and study design. Ref. [16] reported miR-223-3p upregulation in active UC fecal samples, while Ref. [15] found miR-223 upregulation in inactive UC serum samples, suggesting tissue-specific expression patterns. Statistical significance was maintained across studies for the leading candidates, with *p*-values consistently below 0.05 for miR-21, miR-126, and miR-146b-5p in their respective optimal sample types.

### 3.3. Statistical Analysis and Effect Size Evaluation

The cross-study analysis revealed significant heterogeneity in both effect sizes and statistical approaches. Fold changes for upregulated miRNAs ranged from 1.5-fold to 18-fold, with tissue-based studies generally showing higher magnitude changes compared to serum or fecal studies. The highest effect sizes were observed in the [12] study, with miR-126 showing 18-fold upregulation and miR-21 showing 14.7-fold upregulation in active UC tissue samples. Statistical significance was consistently reported for leading candidates, with *p*-values below 0.05 for miR-21, miR-126, and miR-146b-5p across multiple studies. However, several methodological limitations affect the interpretation of these statistical findings. Sample sizes were generally small, in the range of 10–55 participants per disease activity group, limiting statistical power for definitive conclusions. Disease activity assessment methods varied, with studies using different Mayo score cut-offs (≤2 vs. >2, ≤5.27 vs. >5.27) and UCDAI scoring systems, potentially affecting the classification of active versus inactive disease states. Additionally, three studies did not report specific cut-off values for disease activity classification, further complicating cross-study comparisons and meta-analytic approaches.

## 4. Discussion

This systematic review provides compelling evidence that specific miRNAs possess the fundamental characteristics necessary for reliable biomarker development in ulcerative colitis. The convergent findings across seven independent studies, encompassing diverse populations and methodological approaches, suggest that miR-21, miR-126, and miR-146b-5p reflect core pathophysiological processes in active UC rather than random expression variations. The magnitude of expression changes observed (up to 18-fold for miR-126) substantially exceeds typical biological variation and approaches the diagnostic thresholds established for clinically validated biomarkers in other inflammatory conditions. Critically, the tissue specificity of these expression patterns, with minimal changes in inactive UC compared to the healthy controls, indicates that these miRNAs respond specifically to inflammatory activation rather than merely reflecting the presence of underlying chronic disease.

In this paper, we highlighted trends and patterns that result from differential expressions of miRNAs in active ulcerative colitis (aUC) and inactive ulcerative colitis (iUC). In particular, [12] showed a significant upregulation of miR-21 (14.7-fold) and miR-126 (18-fold) in colonic tissues of aUC vs. healthy controls (HCs) In contrast, there were no noticeable alterations in miRNA expression levels when comparing iUC to HCs. These findings highlight the potential role of miR-21 and miR-126 as diagnostic biomarkers specifically for active UC. Indeed, miR-21 has a role in the negative regulation of PDCD4 [19,20], RhoB [21], and NOS-2 induced cellular damage; while its suppression of PTEN led to PTEN repression and an increase in PI3K/Akt activity, cellular pathways may be involved in the development of UC. Thus, the overexpression of miR-21 in macrophages could potentially trigger oxidative stress-induced cellular damage, which could be a contributing factor involved in UC pathogenesis [22,23].

The mechanistic roles of the consistently identified miRNAs provide strong biological plausibility for their biomarker potential and therapeutic relevance. miR-126’s established role in endothelial function regulation and angiogenesis directly correlates with the vascular changes characteristic of active UC, including increased mucosal blood flow, endothelial dysfunction, and aberrant angiogenesis that contributes to the friable, bleeding mucosa observed during active flares. The 18-fold upregulation of miR-126 in active UC tissue likely reflects the intensive vascular remodeling occurring during inflammatory episodes. miR-146b-5p’s involvement in NF-κB signaling pathway regulation positions it as a central coordinator of inflammatory responses in UC. Its consistent upregulation in serum samples across studies suggests systemic inflammatory activation that could serve as a circulating biomarker for disease activity monitoring. The NF-κB pathway’s fundamental role in cytokine production, immune cell activation, and intestinal barrier function makes miR-146b-5p an ideal candidate for reflecting the multifaceted inflammatory processes driving UC pathogenesis.

Study conducted by [13] identified a distinct set of differentially expressed miRNAs in aUC and iUC compared to HCs; such miRNAs include miR-20b (↑ *p* < 0.05) in aUC vs. HCs, miR-125b-1 (↑ *p* < 0.01) in aUC vs. HCs, let-7e (↑ *p* < 0.05) in iUC vs. HCs, and miR-98 (↑ *p* < 0.05) in iUC vs. aUC and HCs [24,25,26,27]. Following up on these results, [14] expanded the panel of miRNAs with similar expression patterns and observed an increase in miR-21-5p, miR-31-5p, and miR-155-5pin aUC compared to HCs [21,22,23,24]. To elucidate the functional relevance of these miRNAs, several have been investigated in the context of specific inflammatory signaling pathways. For instance, let-7e seems to be involved in Toll-like receptor (TLR)-associated signaling pathways, while miR-98 targets pro-inflammatory genes such as MYC and IL-6 [25,26,27,28]. Notably, miR-155 inhibition reduced DSS-induced colonic damage, prevented the development of Th17 cells, and alleviated colitis-associated inflammation by inactivating NF-κB signaling [29,30]. Studies carried out by [16] revealed that miR-199a-5p and miR-223-3p were upregulated in aUC compared to iUC and HCs [30]. Specifically, miR-199a-5p was involved in intestinal barrier dysfunction, whereas the elevated levels of miR-223-3p might contribute to UC pathogenesis by negatively regulating autophagy and IKKα inhibition, which is a potential regulator of inflammation [16,31,32,33].

Interestingly, Ref. [17] reported that miR-146b-5p was overexpressed in serum samples of aUC in comparison to both iUC and HCs. These findings were further supported by [18], who confirmed the potential of miR-146b-5p as a biomarker indicative of active disease. Remarkably, multiple studies have identified an overlap in the overexpression of several miRNAs, including miR-21, miR-223, and miR-146b-5p, suggesting their promise as diagnostic biomarkers for monitoring disease activity in UC.

The consistent overexpression of miR-21, miR-223, and miR-146b-5p across multiple studies highlights their strong potential as diagnostic biomarkers. Moreover, their established roles in inflammatory pathways further support their suitability as targets for clinical research in ulcerative colitis.

### Clinical Implementation Considerations and Diagnostic Utility

From a clinical translation perspective, the identified miRNAs offer several distinct advantages over existing UC biomarkers. The demonstrated stability of these molecules across multiple sample types (tissue, serum, feces) provides flexibility for different clinical scenarios, from routine outpatient monitoring using serum samples to correlation with endoscopic findings using tissue samples. The ability to measure multiple miRNAs simultaneously using standard qPCR platforms available in most clinical laboratories represents a significant advantage over specialized assays required for some emerging biomarkers. However, several challenges must be addressed before clinical implementation. The lack of standardized reference ranges across different populations and the absence of validated cut-off values for clinical decision-making represent significant barriers. The cost-effectiveness compared to fecal calprotectin, currently the most widely adopted non-invasive UC biomarker, requires formal health economic evaluation. Integration into existing clinical workflows and development of point-of-care testing platforms will be essential for widespread adoption. The potential for these miRNAs to complement rather than replace existing diagnostic approaches may provide the most practical implementation pathway. A composite biomarker panel incorporating miRNA measurements with traditional inflammatory markers could potentially improve diagnostic accuracy while maintaining cost-effectiveness. This approach aligns with the growing trend toward precision medicine in inflammatory bowel disease management.

Collectively, these findings emphasize the involvement of specific miRNAs in ulcerative colitis pathogenesis, indicating their potential as both diagnostic and prognostic biomarkers for UC management. The diagnostic utility for assessing current disease activity represents the immediate clinical application supported by current evidence. However, the mechanistic involvement of these miRNAs in inflammatory pathways suggests potential prognostic applications for predicting treatment responses, disease progression, and long-term outcomes. Future investigations involving larger, multicenter cohorts are essential to validate both the diagnostic and prognostic potential of miRNAs. Additionally, head-to-head comparisons with established biomarkers, development of standardized measurement protocols, and health economic evaluations will be crucial for successful clinical translation. The exploration of therapeutic applications targeting these miRNAs could ultimately enhance both the diagnostic precision and treatment effectiveness in ulcerative colitis management.

## Figures and Tables

**Figure 1 ijms-26-06822-f001:**
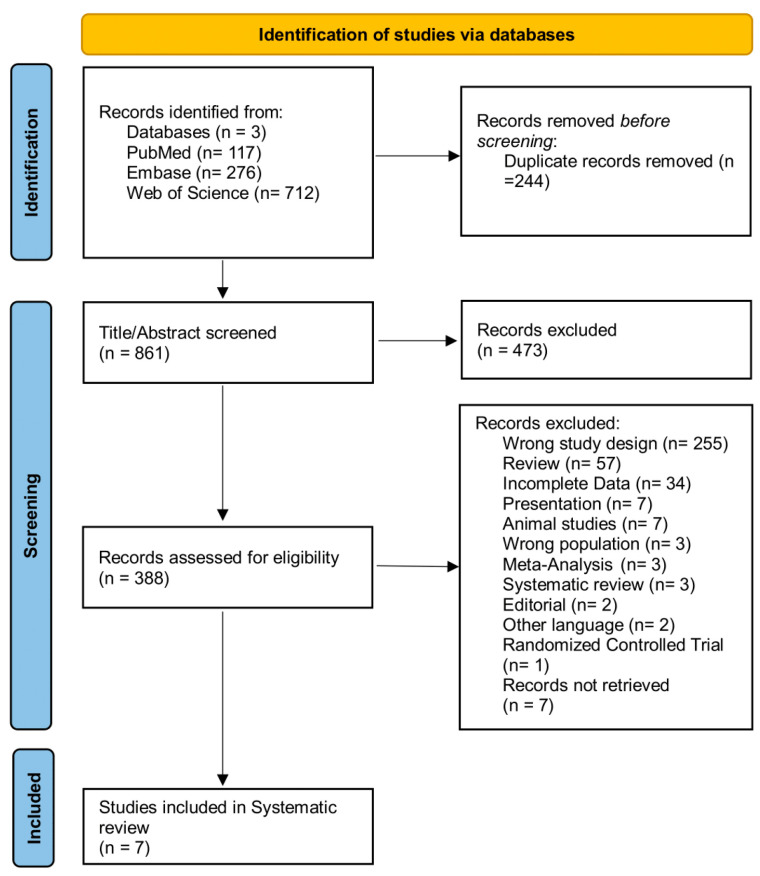
PRISMA flow chart for the search strategy.

**Table 1 ijms-26-06822-t001:** Detailed search strategy.

Database	Keywords	Results
PubMed	(“microRNA” OR “miRNA” OR “mir” OR “microRNAs” OR “miRNAs”)	222,472
	(“biomarker” OR “biomarkers” OR “diagnostic marker” OR “molecular marker”)	927,326
	(“ulcerative colitis” OR “active ulcerative colitis” OR “inactive ulcerative colitis” OR “remission ulcerative colitis” OR “flare ulcerative colitis” OR “quiescent ulcerative colitis”)	66,276
	#1 AND #2 AND #3	117
Web of Science	microRNA OR miRNA OR mir OR microRNAs OR miRNAs AND biomarker OR biomarkers OR diagnostic marker OR molecular marker AND ulcerative colitis OR active ulcerative colitis OR inactive ulcerative colitis OR remission ulcerative colitis OR flare ulcerative colitis OR quiescent ulcerative colitis	712
Embase	microRNA OR miRNA OR mir OR microRNAs OR miRNAs AND biomarker OR biomarkers OR diagnostic marker OR molecular marker AND ulcerative colitis OR active ulcerative colitis OR inactive ulcerative colitis OR remission ulcerative colitis OR flare ulcerative colitis OR quiescent ulcerative colitis	276

**Table 2 ijms-26-06822-t002:** PICO-based inclusion and exclusion criteria.

Parameters	Inclusion Criteria	Exclusion Criteria
Population	Adult individuals with a confirmed diagnosis of ulcerative colitis and differentiation based on disease state	Studies about adult individuals with non-UC conditions (e.g., Crohn’s disease, general IBD, etc.)
Intervention	miRNAs as diagnostic tool to assess UC disease state	Studies that use any other biomolecules as biomarkers
Comparison	N/A	N/A
Outcome	miRNAs as diagnostic biomarker for differentiating active and inactive UC	No significant change in miRNA expression level capable of differentiating active UC from inactive UC

**Table 3 ijms-26-06822-t003:** Quality assessment of the studies based on the Newcastle–Ottawa Scale (NOS).

		NOS Score
Author, Year	Study Design	Selection	Comparability	Outcome/Exposure
[12]	Case–Control	   		 
[13]	Cohort	   		
[14]	Case–Control	   		 
[15]	Cohort	   		
[16]	Case–Control	   		 
[17]	Cohort	   		 
[18]	Case–Control	   		 

**Table 4 ijms-26-06822-t004:** General characteristics of the included studies.

Author and Year	Study Design	Sample Type	Sample Size (aUC/iUC/HC)	Gender (M/F)	miRNA Measurement
[12]	Case–control	Pinch biopsies	12/10/15	5/7, 4/6, 7/8	qRT-PCR
[13]	Cohort	Pinch biopsies	20/19/20	9/11, 6/13, 10/10	miRNA microarray, qPCR
[14]	Case–control	Colonic mucosal biopsies	10/7/10	6/4, 4/3, 5/5	miRNA microarray, qRT-PCR
[15]	Cohort	Serum and fecal samples	10/8/35	4/6, 4/4, 14/21	qPCR
[16]	Case–control	Fecal samples	41/25/66	NR	miRNA microarray, qPCR
[17]	Cohort	Serum samples	55/45/41	NR	qRT-PCR
[18]	Case–control	Blood samples	33/2/30	NR	qPCR

**Table 5 ijms-26-06822-t005:** Summary of differentially expressed circulating miRNAs in ulcerative colitis stratified by disease activity.

Author (Year)	miRNAs Studied	Disease Duration (Years) Mean (Range)	Disease Activity Method	Key Findings
[12]	miR-21, miR-126, miR-375	1.8 (0.5–4) aUC, 4 (2–6) iUC	UCDAI: 0.9 (0–2) iUC, 8.91 (8–10) aUC	miR-126 (↑ 18-fold, *p* < 0.05) in aUC vs. HC. miR-21 (↑ 14.7-fold, *p* < 0.05) in aUC vs. HC. No significant changes in iUC vs. HC.
[13]	miR-20b, -99a, -203, -26b, -98, -125b-1, let-7e	15/5 aUC, 8/11 iUC	Mayo: 0 (0–1) iUC, 6 (2–12) aUC	miR-20b (↑ *p* < 0.05) in aUC vs. HC. let-7e (↑ *p* < 0.05) in iUC vs. HC. miR-98 (↑ *p* < 0.05) in iUC vs. aUC and HC. miR-125b-1 (↑ *p* < 0.01) in aUC vs. HC.
[14]	miR-21-5p, miR-31-5p, miR-146a-5p, miR-155-5p, miR-650, miR-196b-5p, miR-200c-3p, miR-375, miR-200b-3p, miR-422a	7.1 (0.6–20.1) aUC, 6.6 (4.0–14.4) iUC	Mayo: 0 (±0.5) iUC, 8 (±1.5) aUC	miR-21-5p, miR-31-5p, miR-146a-5p, miR-155-5p, miR-650, miR-375 (↑) in aUC vs. HC. miR-196b-5p, miR-196b-3p, miR-200c-3p (↓) in aUC vs. HC.
[15]	miR-16, miR-21, miR-155, miR-223	NR	Mayo: ≤5.27 iUC, >5.27 aUC	miR-21, miR-223 (↑) in iUC vs. HC serum. miR-21, miR-155 (↓) in aUC vs. iUC serum. miR-16, miR-155, miR-223 (↓) in aUC vs. HC feces. miR-155 (↓) in aUC and iUC vs. HC feces.
[16]	miR-199a, miR-223-3p, miR-1226, miR-548ab, miR-515-5p	6.5 aUC, 11.5 iUC	Modified Mayo: ≤2 iUC, >2 aUC	miR-515, miR-548ab, miR-1226 (↓) in aUC vs. HC and iUC vs. HC. miR-199a-5p, miR-223-3p (↑) in aUC vs. HC and iUC.
[17]	miR-197-5p, miR-603, miR-145-3p, miR-574-3p, miR-34a-5p, miR-323a-3p, miR-141-3p, miR-146b-5p, miR-193b-3p, miR-31-5p, miR-27a, miR-27b, miR-944, miR-204-3p, miR-206, miR-24-1-5p, miR-135b-5p	NR	Mayo: ≤2 iUC, >2 aUC	miR-146b-5p (↑) in aUC vs. iUC and HC.
[18]	miR-106a, miR-146b	NR	Mayo: No specific cut values	No significant change between aUC and iUC.

aUC = active ulcerative colitis; iUC = inactive ulcerative colitis; HC = healthy control; UCDAI = Ulcerative Colitis Disease Activity Index; Modified Mayo and Mayo = clinical scoring systems for UC activity; NR = not reported; ↑ = increased expression relative to the comparison group; ↓ = decreased expression relative to the comparison group; *p* < 0.05 and *p* < 0.01 indicate levels of statistical significance reported in the respective studies.

## Data Availability

No new data were created or analyzed in this study. Data sharing is not applicable to this article.

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
