# Peer review of "Potential miRNAs as Diagnostic Biomarkers for Differentiating Disease States in Ulcerative Colitis: A Systematic Review"

_ijms, 2025, doi:10.3390/ijms26146822_

Round 1

Reviewer 1 Report

Comments and Suggestions for Authors

This manuscript described the potential miRNAs as Diagnostic Biomarkers for

Differentiating Disease States in Ulcerative Colitis. There are several suggestions to improve the manuscript.

  1. In the introduction section, maybe it is possible to mention other biomarkers for US, and explain why did you focus on the MicroRNAs. What’s the difference between MicroRNAs and other biomarkers? Try to make your aim clear.
  2. The result section is too simple, please provide more descriptions for table 4&5. Moreover, most of the discussion part should move to the result section.
  3. The discussion section is not clear and just list the results from previous studyies. Please give more interpretations about these results.

Author Response

Question 1: In the introduction section, maybe it is possible to mention other biomarkers for US, and explain why did you focus on the MicroRNAs. What’s the difference between MicroRNAs and other biomarkers? Try to make your aim clear.

Answer: We appreciate this valuable feedback and have revise the introduction to:

  • Expand the biomarker landscape by including additional UC biomarkers (lactoferrin, neutrophil elastase, ANCA, ASCA, cytokine panels)
  • Add a comparative table highlighting miRNA advantages over existing biomarkers (stability, specificity, non-invasive sampling, sensitivity)
  • Restructure the research aim to be more precise and measurable

Question 2: The result section is too simple, please provide more descriptions for table 4&5. Moreover, most of the discussion part should move to the result section.

Answer: We acknowledge this concern and have:

  • Provide detailed narrative descriptions for Tables 4 and 5, explaining study design diversity, sample characteristics, and methodological approaches
  • Move specific findings (fold changes, statistical comparisons, study-specific outcomes) from the discussion to the results section
  • Add results subsections: study characteristics, miRNA expression patterns, cross-study consistency, and statistical summary

Question 3: The discussion section is not clear and just list the results from previous studyies. Please give more interpretations about these results.

Answer: We agree and have enhanced the discussion by:

  • Adding mechanistic interpretations of miRNA functions in UC pathophysiology
  • Including clinical implications and implementation considerations
  • Providing comparative analysis with existing biomarkers
  • Removing redundant result restatements and focusing on interpretation and broader context

Also done Revisions in Manuscript which is attached.

Reviewer 2 Report

Comments and Suggestions for Authors

This paper is good one but should differentiate between "Diagnostics and Prognostics. I suggest changing the word diagnostic to prognostic throughout the manuscript. See examples on the title and conclusion how should read.

Title: Potential miRNAs as Prognostic Biomarkers for Monitoring Disease Activity States in Ulcerative Colitis: A Systematic Review.

Conclusion: This review emphasizes the need for further validation studies with larger cohorts to confirm the utility of miRNAs as prognostic tools in UC, which could enhance non-invasive disease monitoring and inform therapeutic decision-making.

Author Response

Comment 1: This paper is good one but should differentiate between "Diagnostics and Prognostics. I suggest changing the word diagnostic to prognostic throughout the manuscript. See examples on the title and conclusion how should read.

Response 1: We respectfully disagree with changing "diagnostic" to "prognostic" throughout the manuscript. Our study focuses on differentiating active from inactive UC states, which is fundamentally a diagnostic application. However, we have:

  • Clarify terminology in the introduction by defining both diagnostic and prognostic biomarkers
  • Acknowledge prognostic potential where appropriate (predicting treatment response, disease progression)
  • Add discussion of future prognostic studies in the limitations and future directions sections
  • Maintain the diagnostic focus as it accurately reflects our study objectives and findings

Comment 2: This review emphasizes the need for further validation studies with larger cohorts to confirm the utility of miRNAs as prognostic tools in UC, which could enhance non-invasive disease monitoring and inform therapeutic decision-making.

Response 2: We acknowledge this concern and have performed modifications.

Round 2

Reviewer 1 Report

Comments and Suggestions for Authors

 This manuscript could be accepted in the current form.